# The Antioxidant Drug Edaravone Binds to the Aryl Hydrocarbon Receptor (AHR) and Promotes the Downstream Signaling Pathway Activation

**DOI:** 10.3390/biom14040443

**Published:** 2024-04-04

**Authors:** Caterina Veroni, Stefania Olla, Maria Stefania Brignone, Chiara Siguri, Alessia Formato, Manuela Marra, Rosa Manzoli, Maria Carla Macario, Elena Ambrosini, Enrico Moro, Cristina Agresti

**Affiliations:** 1Department of Neuroscience, Istituto Superiore di Sanità, 00161 Rome, Italy; caterina.veroni@iss.it (C.V.); mariastefania.brignone@iss.it (M.S.B.); elena.ambrosini@iss.it (E.A.); 2Institute for Genetic and Biomedical Research (IRGB), The National Research Council (CNR), Monserrato, 09042 Cagliari, Italy; stefania.olla@irgb.cnr.it (S.O.); chiara.siguri@irgb.cnr.it (C.S.); 3Institute of Biochemistry and Cell Biology, IBBC-CNR, Campus Adriano Buzzati Traverso, Monterotondo Scalo, 00015 Rome, Italy; aformato92@gmail.com; 4Core Facilities Technical-Scientific Service, Istituto Superiore di Sanità, 00161 Rome, Italy; manuela.marra@iss.it; 5Department of Molecular Medicine, University of Padova, 35121 Padova, Italy; rosa.manzoli@phd.unipd.it (R.M.); mariacarla.macario@studenti.unipd.it (M.C.M.); 6Department of Biology, University of Padova, 35121 Padova, Italy

**Keywords:** edaravone, aryl hydrocarbon receptor, oligodendrocyte progenitors, zebrafish

## Abstract

A considerable effort has been spent in the past decades to develop targeted therapies for the treatment of demyelinating diseases, such as multiple sclerosis (MS). Among drugs with free radical scavenging activity and oligodendrocyte protecting effects, Edaravone (Radicava) has recently received increasing attention because of being able to enhance remyelination in experimental in vitro and in vivo disease models. While its beneficial effects are greatly supported by experimental evidence, there is a current paucity of information regarding its mechanism of action and main molecular targets. By using high-throughput RNA-seq and biochemical experiments in murine oligodendrocyte progenitors and SH-SY5Y neuroblastoma cells combined with molecular docking and molecular dynamics simulation, we here provide evidence that Edaravone triggers the activation of aryl hydrocarbon receptor (AHR) signaling by eliciting AHR nuclear translocation and the transcriptional-mediated induction of key cytoprotective gene expression. We also show that an Edaravone-dependent AHR signaling transduction occurs in the zebrafish experimental model, associated with a downstream upregulation of the NRF2 signaling pathway. We finally demonstrate that its rapid cytoprotective and antioxidant actions boost increased expression of the promyelinating Olig2 protein as well as of an Olig2:GFP transgene in vivo. We therefore shed light on a still undescribed potential mechanism of action for this drug, providing further support to its therapeutic potential in the context of debilitating demyelinating conditions.

## 1. Introduction

The drug Edaravone (3-methyl-1-phenyl-2-pyrazolin-5-one—EDA) is a small molecule with a high lipid solubility and permeability across the blood–brain barrier that has shown promising neuroprotective activity, particularly in the context of neurological disorders characterized by oxidative stress and neuroinflammation. It was initially approved for the management of ischemic stroke in Japan and later extended to the treatment of amyotrophic lateral sclerosis in Japan, the USA, Canada and Switzerland [1]. EDA has been investigated as a potential treatment in several animal models of central nervous system (CNS) disorders, like multiple sclerosis (MS) [2,3], Parkinson’s disease [4,5], Alzheimer’s disease [6] and traumatic brain injury [7]. In addition, a proof-of-concept study evaluating the protective effect of EDA in patients with early-stage Alzheimer’s disease is ongoing [8].

The clinical efficacy of EDA was primarily linked to its potent scavenging activity against reactive oxygen species (ROS) [9], which thus reduces the oxidative tissue damage that contributes to the initiation and progression of several neurodegenerative diseases [10]. Subsequently, it was shown that EDA’s neuroprotective activity is also driven by the induction of various intracellular signaling pathways. Among these, EDA has been demonstrated to activate the nuclear factor (erythroid-derived 2)-like 2 (NRF2) [2,11,12,13], which regulates the expression of genes encoding phase II detoxification enzymes, contributing to the maintenance of ROS homeostasis. Evidence also shows that EDA exerts an inhibitory effect on the release of pro-inflammatory cytokines by preventing NFκB activation [14]. The neuroprotective activity of EDA has also been extensively linked to the activation of the BDNF-TrkB signaling pathway [15,16,17,18], which contributes to neuronal survival, growth and repair. 

Besides its well-described neuroprotective properties, a significant amount of data provided by us and other groups has shown EDA’s ability to promote remyelination, a neuroprotective, regenerative process aimed at restoring neuronal functions in demyelinating diseases like MS. In particular, these findings demonstrate that EDA promotes the differentiation of oligodendrocytes, the myelin-forming cells of the CNS, and enhances the rate of remyelination in various in vitro and in vivo models of brain damage [3,19,20] in a way that involves the mTORC1 signaling pathway [21].

The neuroprotective properties of EDA appear to be mediated through the activation of various intracellular signaling pathways, in line with the drug’s multifunctional potential. However, there is currently no definitive evidence of a direct interaction between EDA and any of its potential targets. Identifying the biological targets of EDA can contribute to the development of more effective regenerative interventions and provide new insights into the molecular mechanisms of neurodegenerative diseases.

Many different technologies from a wide range of interdisciplinary fields are available to identify the molecular targets of repurposed drugs. Through a computational approach, we tried to identify a potential common target/pathway that could explain the efficacy of various remyelinating drugs, including EDA [19]. Our recent findings indicated that the molecular structure of EDA is not suitable for target identification approaches involving the synthesis of tagged chemical derivatives [22]. In the present study, we employed a transcriptomics-guided drug target discovery strategy, analyzing the expression levels of genes differentially regulated in primary oligodendrocyte progenitor cells (OPCs) exposed or not to EDA, using gene expression data to identify drug-induced protein networks. We found that several transcripts related to the activation of the transcription factor aryl hydrocarbon receptor (AHR) were upregulated in OPCs treated with EDA. The next step involved the evaluation of EDA as a novel AHR agonist by docking and molecular dynamics simulations using an AHR 3D structure, the analysis of AHR nuclear translocation and AHR target gene expression in the human neuroblastoma cell line SH-SY5Y and zebrafish larvae.

## 2. Materials and Methods

### 2.1. Animals

CD1 Swiss mice were purchased from Harlan Laboratories (San Pietro Al Natisone, Udine, Italy). The experimental procedures related to the use of CD1 Swiss mice for the establishment of cell cultures were conducted in accordance with Council Directive 86/609/EC and Decree 116/92 (Authorization n. 87/2017-PR—23 September 2018) issued by the Service for Biotechnology and Animal Welfare of the “Istituto Superiore di Sanità” and by the Italian Ministry of Health. Zebrafish were maintained at 28 °C in 5 L tanks with fish water at neutral pH, according to standard procedures (http://ZFIN.org, accessed on 2 February 2023). All procedures involving zebrafish embryos and larvae were performed according to the Italian Ministry of Health and the Local Institutional Review Board of the University of Padova (OPBA) (protocol code 312/2022-PR of 15 May 2022).

### 2.2. Purified OPC Cultures

OPCs were obtained from neonatal mouse primary mixed glial cultures, as previously described [19,23]. In brief, the forebrains of newborn CD1 Swiss mice were carefully freed of meninges, chopped into 0.2 mm sections and dissociated using a mild trypsinization procedure and gentle mechanical disruption with a Pasteur pipette. Cells were seeded into poly-L-lysine (10 μg/mL, Merck/Sigma-Aldrich, Milan, Italy)-coated 60 mm diameter plastic culture dishes (NUNC, Thermo Fisher Scientific, Waltham, MA, USA) at the density of 1.2 × 10^5^ cells/cm^2^ and grown at 37 °C in a 91.5% air–8.5% CO_2_ humidified atmosphere in Dulbecco’s modified eagle medium (DMEM) containing 10% fetal bovine serum (FBS), 2 mM glutamine, penicillin (50 μg/mL) and streptomycin (50 μg/mL), replacing fresh medium after 1 DIV and every 2–3 days (media, sera and reagents by GIBCO, Thermo Fisher Scientific, Monza, Italy). After 8–10 days, OPCs were detached from the astroglia layer by mechanical dissociation and, to minimize contamination by microglial cells, the detached cell suspension was incubated for 1 h at 37 °C in a 175 cm^2^ culture flask. The non-adhering cells were seeded in the same medium as above at the density of 1 × 10^5^ cells/cm^2^ into poly-L-lysine-coated dishes (96-well plates or in 35 mm diameter plastic culture dishes for the MTT test and real-time RT-PCR assay, respectively). Two hours (h) after plating, the culture medium was replaced with defined serum-free DMEM without thyroid hormones [23]. Macrophage/microglia contamination accounted for less than 1% of the total cells, as assessed by immunostaining with the monoclonal antibody (mAb) CD11b (AbD Serotech, Oxford, UK); glial fibrillary acid protein-positive astrocytes were virtually absent and the majority of cells (>99%) belonged to the oligodendrocyte lineage.

### 2.3. Transcriptome Analysis

Transcriptome analysis was performed at the Next Generation Sequencing area of the Core Facilities Technical-Scientific Service, Istituto Superiore di Sanità, Rome, Italy. Primary OPCs treated with EDA (Merck/Sigma-Aldrich, Italy) 100 µM (*n* = 4) or vehicle alone (*n* = 4) for 14 h were used as the treated and control groups, respectively. Cells were obtained from 4 independent preparations. Targeted transcriptome analysis was performed using the Ion AmpliSeq™ Transcriptome Mouse Gene Expression Kit (Thermo Fisher Scientific, Italy), a targeted gene quantification approach that allows simultaneous gene expression measurement of more than 20,000 mouse RefSeq genes in a single assay. For library preparation, a barcoded cDNA library was first generated with the SuperScript^®^ VILO™ (Thermo Fisher Scientific, Italy) cDNA Synthesis kit from 10 ng of total RNA. Then cDNA was amplified using Ion AmpliSeq™ technology to accurately maintain expression levels of all targeted genes. Amplified cDNA libraries were evaluated for quality and quantified using a Bioanalyzer High Sensitivity Chip (Agilent, Santa Clara, CA, USA). Libraries were then diluted to 100 pM and pooled equally, with eight individual samples per pool. Pooled libraries were amplified using emulsion PCR on Ion Torrent OneTouch2 instruments (OT2) and enriched following the manufacturer’s instructions. Templated libraries were then sequenced on the Ion GeneStudio™ S5 System. AmpliSeq sequencing data were analyzed using the Torrent Suite software version 5.16 and were normalized using reads per million (RPM). Both differential gene expression analysis and principal component analysis were performed using Transcriptome Analysis Console software, version 4.0.2 (TAC, Thermo Fisher Scientific, Italy). Genes showing a differential regulation of ±1.5 and a *p*-value < 0.05 (with ANOVA test) in treated cells compared to control cells were considered for further analysis.

### 2.4. RNA Extraction and Quantitative (q)PCR 

Total RNA was extracted from OPCs, SH-SY5Y cells and zebrafish larvae using a RNeasy mini kit (Qiagen, Redwood City, CA, USA), including a DNase digestion step to eliminate genomic DNA. Five hundred nanograms of RNA were then reverse transcribed using the High Capacity Reverse Transcription kit (Thermo Fisher Scientific). Gene expression analysis was performed via qPCR using the ABI PRISM 7500 System (Applied Biosystem, Thermo Fisher Scientific, Italy), the TaqMan Gene ExpressionMaster Mix (Thermo Fisher Scientific, Italy) and the inventoried FAM-labeled gene expression assays (Thermo Fisher Scientific, Italy) listed in Appendix A. GAPDH was used as a housekeeping gene in all experimental systems (OPCs, SH-SY5Y cells, zebrafish larvae). Gene expression levels were calculated using the formula 2^−ΔΔCt^, where ΔCt is the difference in cycle threshold between the target cDNA and housekeeping cDNA and ΔΔCt is the difference between the ΔCt of treated cells/larvae and the ΔCt of untreated samples.

### 2.5. Preparation of Proteins for Docking 

The structures of AHR were retrieved from the protein data bank (https://www.rcsb.org/ accessed on 15 December 2022), with accession ID 7ZUB [24]. The protein preparation wizard (Schrödinger Suite Release 2022-3) was used to prepare the protein. The bond orders were assigned, and possible missing hydrogen atoms in the 3D structure were added. Epik (Schrödinger Suite Release 2022-3) was employed to generate the heteroatoms’ states at pH 7.4 ± 2.0. Full energetic optimization was performed in the final refinement step using the OPLS4 force field, and the RMSD of heavy atoms was set at 0.3 Å [25].

### 2.6. Preparation of Ligands for Docking 

The structures of all ligands were prepared with LigPrep (Schrödinger Suite Release 2022-3) using the OPLS4 force field, generating the possible ionization states at pH 7.0 ± 2.0 and retaining the specified chirality. 

### 2.7. Docking Studies 

The 3D structure includes AHR-HSP90-XAP2, with the ligand indirubin (INDI) bound to the PSA-B domain of AHR [24]. Docking was performed on the entire protein and focused on the INDI binding site. The Receptor Grid Generator was employed to generate suitable grids for the docking with Glide [26,27]. Two grids were generated, one encompassing the entire protein domain and the other, with more restricted dimensions of 46 × 46 × 46 Å, utilizing the INDI center in the domain as its grid center; the chosen force field was OPLS_2005 [28]. Glide-XP (Schrödinger Suite Release 2022-3) [26,27,29] was chosen as one of the docking protocols. Three poses per ligand were kept during the post-docking minimization using a threshold of 0.50 kcal/mol and, also in this case, the OPLS_2005 was used as the force field. The results from docking were then submitted to MM-GBSA (Molecular Mechanics with Generalized Born and Surface Area solvation) [30] using VSGB as the solvation model and OPLS4 as the force field [31]. Using AutoDock 4 software [32], Gasteiger charges [33] were assigned to the protein structure and again two grids were generated with AutoGrid [34]. The established dimensions were 50 × 50 × 50 Å entered within the binding site for the focused one, whereas the other one encompassed the entire protein (blind). Docking experiments were performed using the genetic algorithm [35,36] with 250 trials and a population of 500 individuals. The maximum number of generations and evaluations was set to 10,000,000 and 25,000,000, respectively. The other parameters were kept as defaults.

### 2.8. Molecular Dynamics

Molecular dynamics (MD) simulations were performed using Desmond (Schrödinger Suite Release 2022-3) [37] and the TIP3P solvent model [38] was employed. The ligand–receptor complex was placed in an orthorhombic water box, which extended 10.0 Å, and the box volumes were minimized and neutralized by adding ions (Na^+^ or Cl^−^). The OPLS4 force field was chosen. MD simulations were conducted for a duration of 500 ns in the NPT ensemble, with the maintenance of a constant temperature (300.0 K) using the Nosé–Hoover thermostat [39], while the Martyna–Tobias–Klein barostat method [40] was used for the pressure (1.01325 bar). Generated trajectories were subjected to clustering based on RMSD using Schrödinger’s trj_cluster.py script [41] and subsequently analyzed through MMGBSA analysis using the thermal mmgbsa.py script integrated within Desmond [37].

### 2.9. SH-SY5Y Cell Cultures and Treatments 

The SH-SY5Y cell line was kindly provided by Dr. Cinzia Mallozzi (ISS, Rome, Italy) [42] and maintained in culture in Dulbecco’s modified Eagle medium (DMEM)/nutrient mixture F-12 (Merk/Sigma-Aldrich, Italy) supplemented with 10% FBS (GIBCO Life Technologies, Grand Island, NY, USA), 1% Glutamine and 1% Penicillin–Streptomycin (Merck/Sigma-Aldrich, Italy) at 37° C in a humidified incubator with 5% CO_2_. To study AHR nuclear translocation, cells were plated in 100 mm diameter dishes (1 × 10^6^ cells), maintained in culture conditions for 48 h, and stimulated for different time lengths (15 min (min), 30 min, 2 h, 6 h) with 100 µM of EDA (Merk/Sigma-Aldrich, Italy) or 1 µM INDI (Merck/Sigma-Aldrich, Italy). For CYP1a1 and NRF2 protein expression analysis, cells were treated for 24 h with EDA 100 µM or INDI 1 µM. To inhibit the AHR nuclear translocation, cells were treated with 1 µM of the AHR antagonist III GNF351 (Merck/Sigma-Aldrich, Italy) for 15 min before the addition of EDA or vehicle alone (DMSO).

### 2.10. Protein Extract Preparation and Western Blotting 

Cytosolic and nuclear protein extracts from the SH-SY5Y cell line, either untreated or treated for AHR nuclear translocation or AHR inhibition experiments, were obtained using a Nuclear Extraction Kit (#ab113474; Abcam, Milan, Italy), as outlined in the manufacturer’s protocol. Briefly, cell samples were washed in ice-cold phosphate-buffered saline (PBS) and centrifuged for 5 min at 1000 rpm. Then, cells were resuspended in an extraction buffer on ice for 10 min and centrifuged for 1 min at 12,000 rpm. After centrifugation, the cytosolic and nuclear fractions were collected and stored at −80 °C for Western blot analysis. Quantification of protein loading content was carried out using a bicinchoninic acid assay (BCA) protein assay kit (Thermo Fisher Scientific, Italy). Equal amounts of proteins (40 μg) were resolved on SDS–PAGE using gradient (4–12%) pre-casted gels (ThermoFisher Scientific, Italy) and transferred onto nitrocellulose or PVDF membranes using the Trans-Blot Turbo Transfer System (BioRad, Hercules, CA, USA). Membranes were blotted overnight (ON) at 4 °C using anti-AHR mAb (1:1000, Santa Cruz Biotechnology, Paso Robles, CA, USA), anti-CYP1A1 mAb (1:200, Santa Cruz Biotechnology, CA, USA), anti-NRF2 mAb 1:500 (Santa Cruz Biotechnology, CA, USA), anti-GAPDH mAb (1:1000, Santa Cruz Biotechnology, CA, USA), anti-Actin mAb (1:2000, Santa Cruz Biotechnology, CA, USA), anti-Lamin B1 mAb (1:1000, Santa Cruz Biotechnology, CA, USA) or anti-AHRR mAb (1:300, Santa Cruz Biotechnology, CA, USA). After repeatedly washing in Tris-buffered saline (TBS), membranes were incubated with horseradish peroxidase-conjugated anti-mouse Ab (1:5000; BioRad Laboratories, Segrate, Milan, Italy) for 1 h at RT. Immunoreactive bands were visualized using an enhanced chemiluminescence reagent (Thermo Fisher Scientific, Italy) and exposed on a BioRad ChemiDoc XRS system. Densitometric analyses of Western blot experiments were performed using NIH ImageJ v 1.53 software (https://imagej.net/ij/, accessed on 14 March 2023) or the BioRad ChemiDoc XRS system. 

### 2.11. Drug Treatments on Fish

Wild-type and transgenic embryos were subjected to drug exposure at 8 h post-fertilization (hpf). The chorion of each single embryo was manually perforated with a small needle before exposure to each treatment. EDA and GNF351 were dissolved in fish water at the reported concentrations, changing the medium after 24 h in the two-day treatments. After the treatments, larvae were euthanized with an overdose of Tricaine and their trunks were manually dissected using needles. After several washes in PBS, pooled trunk tissues were solubilized in Tissue Extraction Buffer (Thermofisher, Italy) containing protease and phosphatase inhibitors (Thermo Fisher Scientific, Italy). For fish transiently expressing the XRE-reporter transgene, we first removed the luciferase coding sequence from the PXRE3G5-FL plasmid [43] and cloned the eGFP coding sequence via HindIII and EcoRI digestion and ligation. We next microinjected one-cell-stage embryos with 500 pg/embryo and proceeded with the treatment as described above. 

### 2.12. Statistical Analysis 

Statistical analyses were performed using IBM SPSS statistics 26.0 software. A two-way ANOVA test for repeated measures was applied for comparisons over time, while unpaired Student’s *t*-tests were used for comparisons between two groups. Results are expressed as mean ± standard error of the mean (SEM). *p* values of less than 0.05 were considered statistically significant and are expressed as * for *p* < 0.05, ** for *p* < 0.01 and *** for *p* < 0.001.

## 3. Results

### 3.1. Edaravone Increases the Expression of AHR-Related Target Genes in Primary Mouse OPCs

Targeted transcriptome analysis was performed to analyze the genes and pathways that were differentially regulated in primary OPCs with or without EDA treatment (100 μM, 14 h). The incubation period was chosen based on the results obtained in preliminary experiments, which showed that shorter incubation times (2–8 h) were not sufficient to induce a substantial modulation of gene expression. As shown in Figure 1, 1132 genes were significantly modulated by EDA treatment compared to control samples (ANOVA, *p* < 0.05). 

Among these, 249 genes with a fold change range of ±1.5-fold of the mean reads assigned per million mapped reads (RPM) values between EDA-treated and control samples were selected for further analysis. Raw transcriptomics data are supplied as Appendix A. Gene function was assigned using the Database for Annotation, Visualization and Integrated Discovery (DAVID, NIH) [http://david.abcc.ncifcrf.gov/ (accessed on 15 September 2022)]. Table 1 displays the functional classification of the significantly up-regulated (*n* = 57) and down-regulated (*n* = 192) genes in biological pathways.

The analysis revealed that EDA treatment significantly enhanced the expression of three genes involved in cytochrome p450 (CYP) activity: aryl-hydrocarbon receptor repressor (*Ahrr*), cytochrome P450 family 1 subfamily A member 1 (*Cyp1a1*) and B member 1 (*Cyp1b1*). All these genes are known as key targets of the AHR pathway, as *Cyp1a* and *Cyp1b* involved in the cellular detoxification response [44]. We validated this finding through additional experiments performed by qPCR, which demonstrated a significant increase in the expression levels of *Ahrr*, *Cyp1a1* and *Cyp1b1* after treatment of OPCs with EDA at concentrations of 30 μM and 100 μM (Figure 2). 

Collectively, we could infer that among different primary targets, EDA is responsible for AHR pathway activation in mouse OPCs. 

### 3.2. Edaravone Is Predicted to Be an AHR Ligand

To verify the hypothesis that EDA activates the AHR signaling pathway by directly binding to AHR, we investigated the potential binding mode through docking studies using INDI and leflunomide, known AHR agonists, as reference compounds. The AutoDock 4 and Glide software tools [29,32] were used to carry out both focused and blind docking for all compounds, leveraging the cryo-EM structure that was recently published [24]. Next, the best docking poses of Glide complexes were chosen to perform binding energy calculations using the MM-GBSA protocol. The MM-GBSA rescoring analysis was carried out to eliminate false positive predictions. The results of these analyses consistently indicated that EDA, along with the two reference compounds, binds to AHR at the same site as the complexed INDI (Figure 3). 

In particular, the two software tools identified identical orientations for INDI and EDA, except for the orientation of the benzene ring in EDA. In contrast, the two software poses for leflunomide docked it within the binding pocket but with different orientations. As shown in Table 2, EDA exhibits higher docking energies (−7.55 kcal/mol in Glide and −5.97 kcal/mol in AutoDock 4) and binding free energy (dG bind, −45.03 kcal/mol) than the two agonists, yet these values are still within satisfactory ranges.

To assess the stability of the AHR-EDA complex, a MD study was conducted for 500 ns, employing the docking-derived binding pose from the Glide software as the starting input. The dynamics confirmed the binding between EDA and AHR but unveiled that EDA frequently undergoes binding transitions within the pocket, shifting slightly from the binding identified by docking (Appendix A).

### 3.3. Edaravone Induces AHR Nuclear Translocation and AHR Target Gene Expression in the SH-SY5Y Neuroblastoma Cell Line 

To validate the docking prediction and assess whether EDA-mediated AHR pathway induction could be conserved in a human experimental model, we assessed the ability of EDA to induce the nuclear translocation of AHR and subsequent expression of endogenous AHR target genes in the neuroblastoma cell line SH-SY5Y, which represents a relevant cellular model for investigating this signaling pathway [45]. Cells were treated with 100 μM EDA for 15 min, 30 min, 2 h and 6 h. Cell lysates were then collected and subjected to fractionation into cytosolic and nuclear fractions. The Western blot results showed that the AHR protein levels significantly decreased in cytosolic-containing fractions within 2 h of EDA treatment, while increasing AHR protein levels were detected in the nuclear fractions over 6 h of EDA treatment (Figure 4). 

In addition, the expression of the *AHRR* and *CYP1A1* genes was examined at both the transcript and protein levels. SH-SY5Y cells were incubated with 30 and 100 μM EDA for 14 h, using INDI, the known AHR endogenous ligand, as positive control.

EDA significantly increased the *AHRR* and *CYP1A1* transcript levels (Figure 5a), as well as their protein levels (Figure 5b,c). As NRF2 is a key downstream target of AHR [46], we next evaluated EDA activity on *NRF2* expression in our experimental model. The Western blot data show a significant up-regulation of NRF2 protein expression after treatment of SH-SY5Y with EDA at a concentration of 100 µM for 24 h compared to unstimulated cells (Figure 5d). Our findings demonstrated that, in response to EDA, AHR is activated and translocates from the cytoplasm to the nucleus, where it induces the expression of its target genes.

### 3.4. Edaravone Promotes Activation of the AHR and NRF2 Pathways and Olig2 Transgene Expression in Zebrafish Larvae

To confirm EDA activity on the AHR pathway in an in vivo model, we measured the expression levels of the *cyp1a1* zebrafish orthologue and the two AHRR zebrafish genes (*ahrra* and *ahrrb*) in EDA-treated larvae. Eight hpf embryos were exposed to EDA at 10 and 30 μM or DMSO for 24 and 48 h and the *cyp1a*, *ahrra* and *ahrrb* transcript levels were determined via qPCR. As shown in Figure 6a, EDA induced a significant up-regulation of *cyp1a* in treated larvae. To further confirm that EDA was specifically inducing the AHR pathway at a transcriptional level, we transiently overexpressed a plasmid containing three xenobiotic responsive elements (XRE) upstream of the eGFP coding sequence [43]. In particular, one-cell-stage embryos were microinjected with the XRE-eGFP-containing plasmid and subjected to 24 h of treatment with EDA or DMSO. 

As shown in Appendix A, we detected an increased number of GFP fluorescent cells in EDA-treated microinjected fish when compared to the number in microinjected controls. 

To further investigate and corroborate the antioxidant response elicited in vivo by EDA, we first treated a recently generated Nrf2 pathway reporter fish [47] with EDA for 48 h and evaluated the expression levels of the reporter gene (GFP) via Western blot. Compared to age-matched DMSO-treated fish, EDA-treated fish exhibited higher (although at the margin of statistical significance (*p* = 0.07) GFP protein levels (Figure 6b). We next evaluated in the same EDA-treated fish and DMSO controls the expression levels of the transcription factor Nrf2 and the glutamate cysteine ligase catalytic subunit (Gclc), which is the rate-limiting enzyme in the synthesis of glutathione and a NRF2 downstream target [48]. As shown in Figure 6c,d, the protein levels of both Nrf2 and Gclc were significantly upregulated in EDA-treated fish when compared to age-matched controls. As accumulating evidence indicates that AHR and NRF2 are involved in oligodendrocyte development and myelination processes [49,50], we also analyzed the effects of EDA on the induction of oligodendrocyte lineage specification using the previously described *Tg(Olig2:eGFP)^vu12^* line [51]. 

As shown in Figure 7, treatment of 8 hpf *Tg(Olig2:eGFP)^vu12^* transgenic fish with 30 μM EDA for 48 h induced a significant increase in reporter protein expression (GFP). Notably, the increased transgene expression detected in EDA-treated larvae was paralleled by elevated Olig2 protein levels in EDA-treated fish lysates when compared to those of age-matched controls (Appendix A).

Collectively, these results confirmed that in vivo EDA treatment triggers the activation of the AHR and NRF2 signaling axis and fosters Olig2+ oligodendrocyte lineage specification, likely because of increased Olig2 protein levels. 

### 3.5. Edaravone-Mediated Induction of CYP Genes Is Dampened by the AHR Antagonist GNF-351 in SH-SY5Y Cells and Zebrafish

We next verified whether the up-regulation of genes associated with the AHR pathway could be prevented by the administration of the competitive AHR antagonist GNF-351, which exhibits effective antagonism against a wide range of AHR ligands [52]. SH-SY5Y cells were treated with EDA (30 µM and 100 µM) in the presence or absence of 1 µM GNF-351 for 14 h. The dose of 1 µM was selected based on preliminary dose–response experiments. As shown in Figure 8a, co-treatment with GNF-351 completely prevented the EDA-dependent increase in *AHRR* and *CYP1A1* transcript levels. In agreement with these observations, we also co-treated fish larvae with 30 µM EDA and 5 µM GNF-351 for 24 h and evaluated the expression levels of the target genes *cyp1a, ahrra* and *ahrrb*. Figure 8b shows that the inhibition of AHR by GNF-351 was able to prevent the EDA-dependent upregulation of the target genes *cyp1a* and *ahrr*. Based on these findings, we can state that the upregulation of AHR target genes is directly mediated by the impact of EDA on AHR activity.

### 3.6. GNF-351 Competes with Edaravone for the Same AHR Binding Site 

Next, we wanted to assess whether EDA and GNF-351 can efficiently and directly interact with the same ligand binding pocket of AHR through docking and molecular dynamics studies. Both AutoDock 4 and Glide confirmed binding of GNF-351 in the same pocket as EDA, but with a lower energy (−8.16 kcal/mol and −10.55 kcal/mol, respectively), confirming the higher activity and affinity of the antagonist. The two software packages identified similar interactions, including pi-pi stacking with His 291 and Phe 324, aromatic H-bond with Ser 346 and pi-pi stacking with Phe 295 for AutoDock 4 and aromatic H-bond with Ser 320 for Glide (Appendix A). To assess binding stability, MD was performed, confirming GNF-351’s stable binding to AHR. Throughout more than 30% of the dynamics, H-bond interactions were observed with Ser 365 (95%), Phe 295 (42%) and Tyr 322 (30%), along with pi-pi stacking with Tyr 322 (76%), Phe 295 (50%) and His 291 (34%) (Appendix A). During simulation, GNF-351 exhibits stabilization within the pocket and undergoes movement relative to the identified docking (Appendix A). The average MMGBSA calculation throughout the dynamic is −95.209 ± 5.106 kcal/mol, once more demonstrating a lower value compared to EDA. This further confirms the higher affinity of GNF-351 for AHR within the identical pocket occupied by EDA.

## 4. Discussion

EDA is a free radical scavenger and antioxidant agent with neuroprotective and remyelinating properties. Uncovering direct molecular targets that mediate its biological activity is critical to understanding the full therapeutic potential of the drug. 

By performing in vitro, in vivo and in silico experiments, our current study establishes that EDA is a novel agonist of the transcription factor AHR and induces an AHR-dependent expression of known target genes. 

AHR was first characterized as a ligand-induced transcriptional regulator involved in the adaptive response for xenobiotic detoxification [53]. Accumulating evidence strongly supports AHR’s relevant role in an array of physiological processes, like cellular homeostasis, cell development and immune response [54]. AHR is activated by environmental contaminants, naturally occurring compounds and endogenous metabolites. Following ligand binding, AHR translocates into the nucleus, forms a dimer with the nuclear translocator ARNT and stimulates the transcription of target genes carrying xenobiotic responsive elements (XREs) in the promoter region, such as CYP1 family genes and the repressor AHRR, which counteracts AHR-dependent gene expression. 

Using targeted transcriptomic analysis and qPCR, we observed a significant increase in the expression of genes related to the AHR pathway (*CYP1A1, CYP1B1, AHRR*) in mouse OPCs and human neuroblastoma SH-SY5Y cells after treatment with EDA. Additionally, we showed that EDA was able to promote the expression of AHR target genes and induce reporter activity in transient XRE:eGFP overexpressing zebrafish larvae. 

*CYP1A1* gene expression is primarily regulated by the AHR, thus establishing this gene as a distinctive marker of AHR pathway activation [55]. The complete inhibition of *CYP1A1* induction in neuroblastoma cells and *cyp1a* in zebrafish by the AHR antagonist GNF-351 strongly supports the hypothesis that AHR activation is instrumental for EDA-induced CYP pathway stimulation. 

In support of the assumption that EDA acts as an AHR ligand, our in silico studies predicted a favorable and stable energy profile for the drug within the binding pocket over time. The evidence that EDA and GNF-351 bind to the same AHR pocket suggests a competitive antagonism between the two ligands. Notably, GNF-351 has an advantage in this competition due to its higher binding affinity compared to EDA, as also pointed out. The finding that EDA promoted AHR nuclear translocation in SH-SY5Y cells reinforces the idea that AHR activation may occur in the presence of direct ligand binding, excluding the non-genomic mechanisms previously reported for some compounds in the activation of AHR target genes [56]. 

Our research also showed that EDA effectively enhances *NRF2* expression in both SH-SY5Y cells and zebrafish larvae. This result supports the involvement of NRF2 signaling in the drug’s antioxidant activity, as previously demonstrated in various models of neurodegenerative diseases [2,11,12,13]. Given that *NRF2* is a target gene of AHR, bearing at least one functional XRE sequence in its promoter [46], and is also activated through ROS generated by CYP1A1 [57], we postulate that EDA’s activity is possibly mediated through the AHR-NRF2 pathway. The complex crosstalk between these two signaling pathways leads to the induction of cytoprotective genes encoding detoxificating and antioxidant enzymes that may explain many of the effects already described for the drug [58].

We observed that EDA activates the AHR pathway during the differentiation of purified mouse OPCs in vitro and in developmental oligodendrogenesis in zebrafish (24–56 hpf). We also showed that in zebrafish larvae EDA not only activates the AHR-NRF2 pathway but also increases Olig2 protein levels and *Olig2:GFP* transgene expression. This aligns with recent findings indicating that proper modulation of the AHR signal is essential for oligodendrocyte development in zebrafish models [59], although, at odds with this work, we found that AHR-NRF2 pathway activation by EDA increases reporter expression in the *Olig2:GFP* transgenic line. The apparent contrasting effects reported by Martins and colleagues on AHR pathway induction and oligodendrogenesis may be ascribed to additional secondary effects produced by tetrachlorodibenzo-para-dioxin when compared to those of EDA. Alternatively, underexplored mechanisms of EDA action may be dominant over the previously described negative effect of AHR activation on the oligodendroglial population expansion. To support the first scenario, the key role of AHR in oligodendrocyte differentiation and myelination was already elucidated through the analysis of AHR knockout models [49,60] and subsequently strengthened by the finding that AHR activation increases sphingolipid levels and axon myelination [61]. Therefore, the combination of our data with evidence from the literature leads us to suggest AHR as the target responsible for the pro-myelinating effect of EDA [3,19,20,21], likely due to the expansion of the oligodendroglial lineage.

Ensuring the proper modulation of AHR signaling is crucial for maintaining cellular homeostasis. The inactivation or overactivation of the AHR pathway has been demonstrated to contribute to the dysregulation of proinflammatory and neurodegenerative mechanisms in several neurological diseases [62]. Notably, a recent study by Tsaktanis et al. [63] found a decrease in AHR agonistic activity in the serum of MS patients, showing a correlation with disease progression. 

EDA, along with other drugs already in use in the clinic [64,65], emerges as an ideal AHR agonist, as it triggers the favorable aspects of AHR activation without the undesired side effects observed with dioxin-like chemical pollutant derivatives. While recognizing the need for further studies to establish the mechanistic link between AHR activation and NRF2 pathway induction, as well as its correlation with increased expression of the downstream *Olig2* target, we envisage that the identification of AHR as a key molecular target of EDA will pave the way for more informed design of new molecules with improved AHR binding activity and affinity, which might be considered for the screening of pro-myelinating compounds.

## Figures and Tables

**Figure 1 biomolecules-14-00443-f001:**
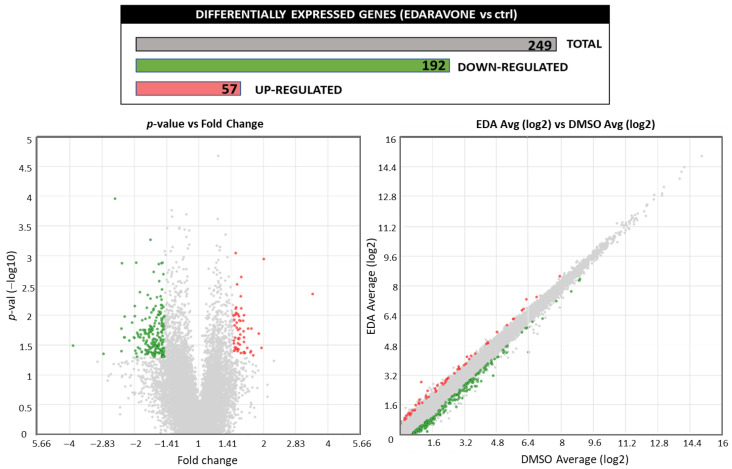
Effect of EDA treatment on the OPC transcriptome. Purified OPCs were incubated with 100 µM EDA or vehicle alone (DMSO) for 14 h. RNA was extracted, reverse transcribed and subjected to targeted transcriptome analysis. Treatment with EDA regulated the expression of 249 genes ranging in a ±1.5-fold change with a *p*-value < 0.05. The volcano plot shows statistical significance (*p*-value) versus the magnitude of change (fold change); red and green dots represent the up- and down-regulated genes, respectively. The image was edited using BioRender.com (https://www.biorender.com/ accessed on 3 April 2024).

**Figure 2 biomolecules-14-00443-f002:**
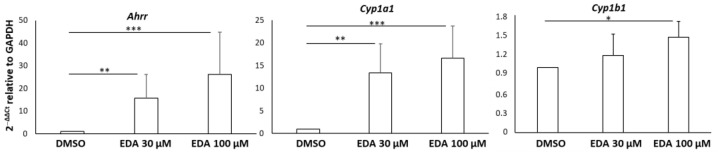
Validation of the effect of EDA treatment on AHR-related transcript expression in OPCs. OPCs were treated with EDA 30 µM, 100 µM or vehicle (DMSO) alone for 14 h. Total RNA was extracted and reverse transcribed and then the expression of the selected genes was evaluated using qPCR. Data are expressed as 2^−ΔΔCt^ relative to the housekeeping gene *Gapdh*. Bars represent the mean ± SEM of 5 independent experiments. * *p* < 0.05, ** *p* < 0.01 and *** *p* < 0.001 with unpaired Student’s *t*-test. The image was edited using BioRender.com (https://www.biorender.com/ accessed on 3 April 2024).

**Figure 3 biomolecules-14-00443-f003:**
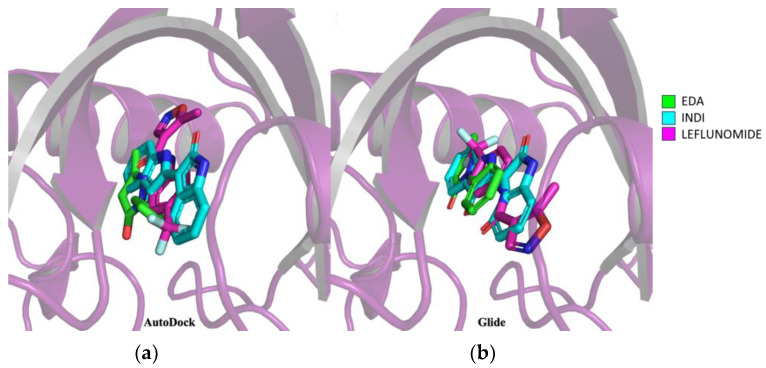
Prediction of EDA–AHR binding mode by molecular docking. (**a**) Superimposition of docking results on AHR of EDA in green, INDI in cyan and leflunomide in magenta using Autodock 4 software. (**b**) Superimposition of docking results on AHR of EDA in green, INDI in cyan and leflunomide in magenta using Glide software. In the 3D structures of the docked compounds, the oxygen atom is represented in red, nitrogen in blue while the fluorine atoms of leflunomide are represented in light blue. The image was edited using BioRender.com (https://www.biorender.com/ accessed on 3 April 2024).

**Figure 4 biomolecules-14-00443-f004:**
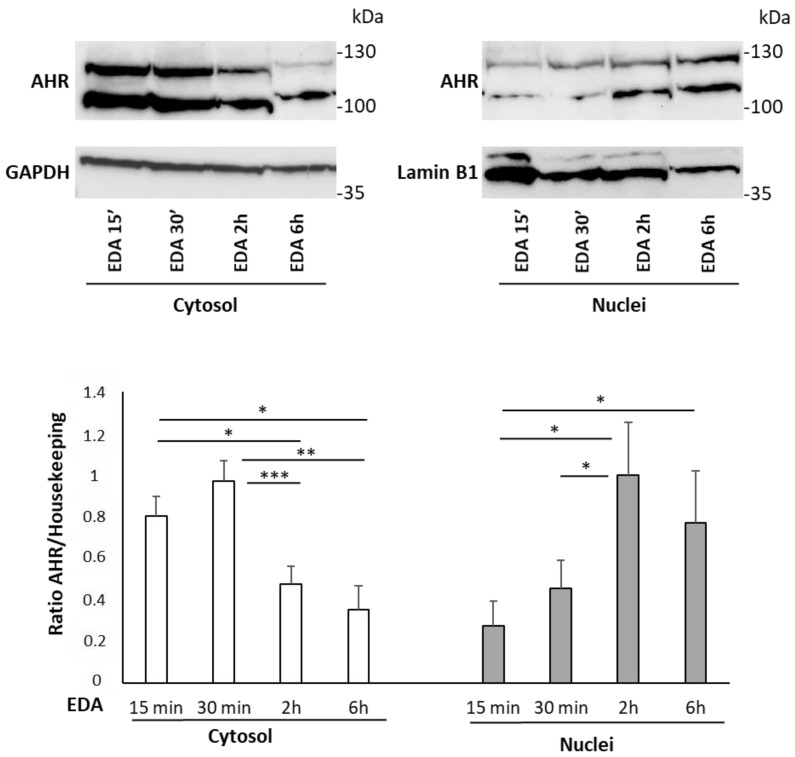
EDA induction of AHR nuclear translocation in the SH-SY5Y neuroblastoma cell line. SH-SY5Y human neuroblastoma cells were incubated with 100 µM EDA for 15 min, 30 min, 2 h and 6 h. The cytosolic and nuclear fractions were separated and the expression level of AHR in each fraction was evaluated by Western blot analysis. GAPDH and LAMINB1 were used for protein content normalization in cytosol and nuclei, respectively. Bars represent the mean ± SEM of 4 experiments. * *p* < 0.05, ** *p* < 0.01 and *** *p* < 0.001 by 2-way ANOVA analysis for repeated measures. For AHR protein quantification, the higher MW band has been considered. The image was edited using BioRender.com (https://www.biorender.com/ accessed on 3 April 2024).

**Figure 5 biomolecules-14-00443-f005:**
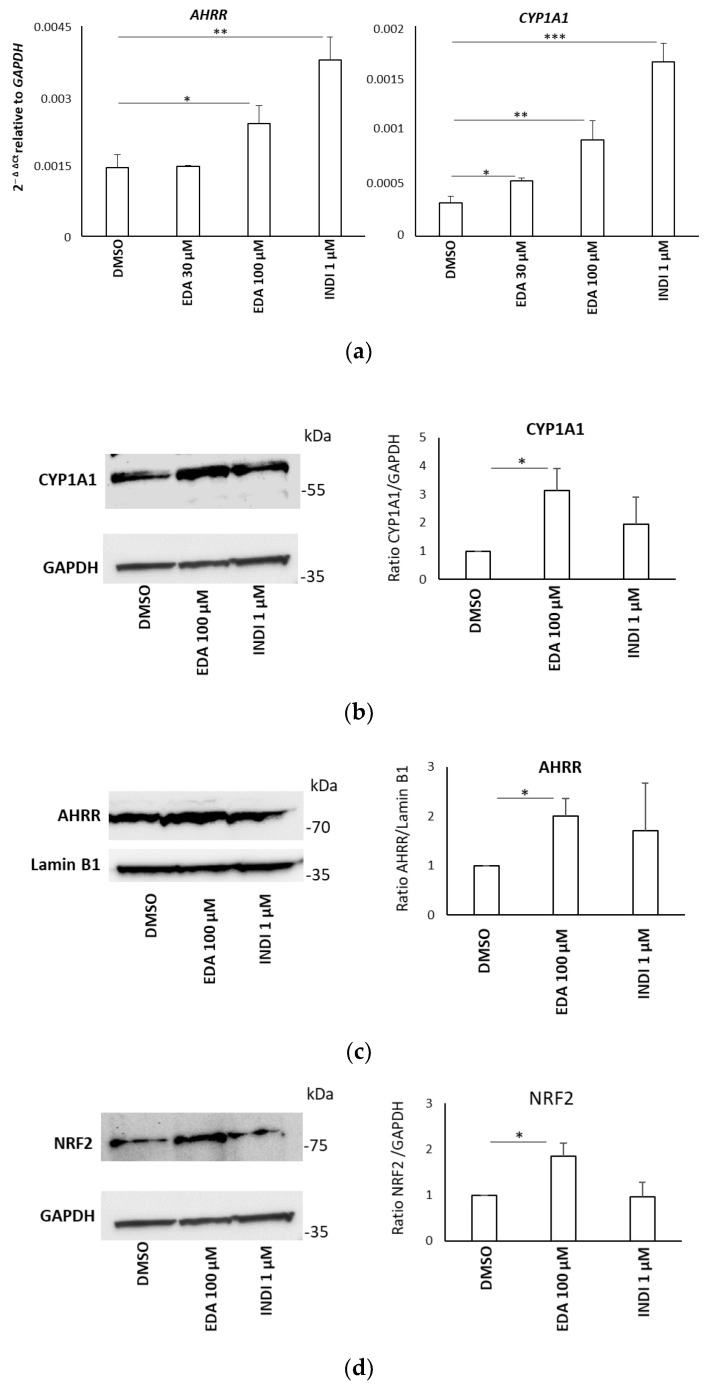
EDA induction of AHR target genes in the SH-SY5Y neuroblastoma cell line. (**a**) SH-SY5Y cells were incubated with 30 µM EDA, 100 µM EDA, 1 µM INDI or DMSO alone for 14 h. Total RNA was extracted and the expression of *AHRR* and *CYP1A1* transcripts was evaluated using qPCR. Data are expressed as 2^−ΔCt^ relative to the housekeeping gene *GAPDH.* (**b**–**d**). SH-SY5Y cells were treated with 100 µM EDA, 1 µM INDI or DMSO alone for 24 h and CYP1A1 (**b**), AHRR (**c**) and NRF2 (**d**) protein expression was investigated by Western blot. Data are expressed as the ratio between AHR and the GAPDH reference. Bars represent the mean ± SEM of 3 experiments. * *p* < 0.05, ** *p* < 0.01 and *** *p* < 0.001 using unpaired Student’s *t*-test. The image was edited using BioRender.com (https://www.biorender.com/ accessed on 3 April 2024).

**Figure 6 biomolecules-14-00443-f006:**
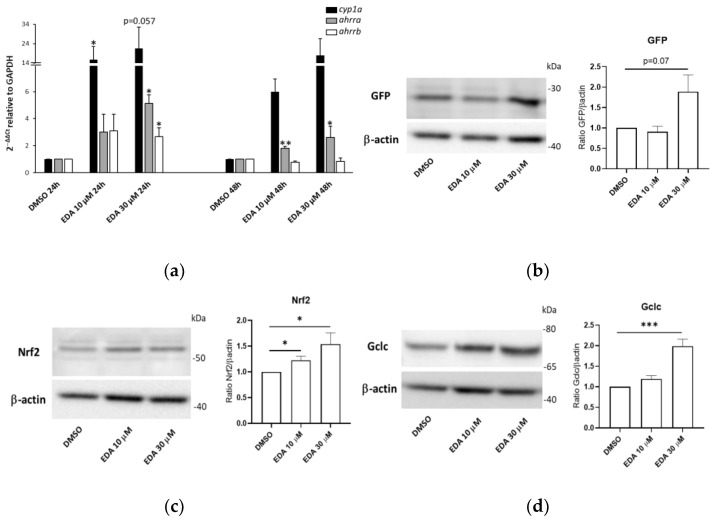
EDA promotes AHR and NRF2 pathway activation in zebrafish larvae. (**a**) *cyp1a, ahrra* and *ahrrb* transcript expression in zebrafish larvae at 56 hpf treated with vehicle (DMSO) or 10 or 30 µM EDA for 24 and 48 h. Asterisks above bars indicate statistically significant changes compared to DMSO-treated (control) samples. (**b**) Representative Western blot for the eGFP reporter protein on fish trunk whole lysates from control DMSO and EDA-treated *Tg(8x AORE:EGFP)^ia201^* larvae at 56 hpf. Fish were treated for 48 consecutive hours. For both gene expression and Western blot analysis, data are expressed as the mean ±SEM of 4 biological replicates (10 larvae per replicate). (**c**,**d**) Representative Western blot for Nrf2 and Gclc proteins on fish trunk whole lysates from control DMSO and EDA-treated larvae at 56 hpf. Data are expressed as the mean ± SEM of 6 biological replicates (10 larvae per replicate). * *p* < 0.05, ** *p* < 0.01 and *** *p* < 0.001 with unpaired Student’s *t*-test. The image was edited using BioRender.com (https://www.biorender.com/ accessed on 3 April 2024).

**Figure 7 biomolecules-14-00443-f007:**
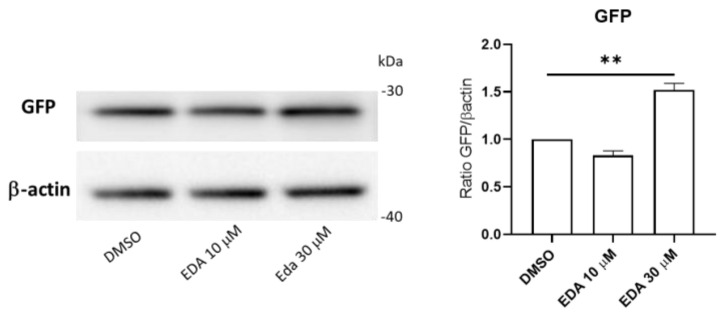
EDA treatment induces reporter expression in Olig2 transgenic fish. Representative Western blot for the eGFP reporter protein on fish trunk whole lysates from control DMSO and EDA-treated *Tg(Olig2:eGFP)^vu12^* transgenic fish. Data are expressed as the mean ± SEM of 3 biological replicates (10 larvae per replicate). ** *p* < 0.01 with unpaired Student’s *t*-test. The image was edited using BioRender.com (https://www.biorender.com/ accessed on 3 April 2024).

**Figure 8 biomolecules-14-00443-f008:**
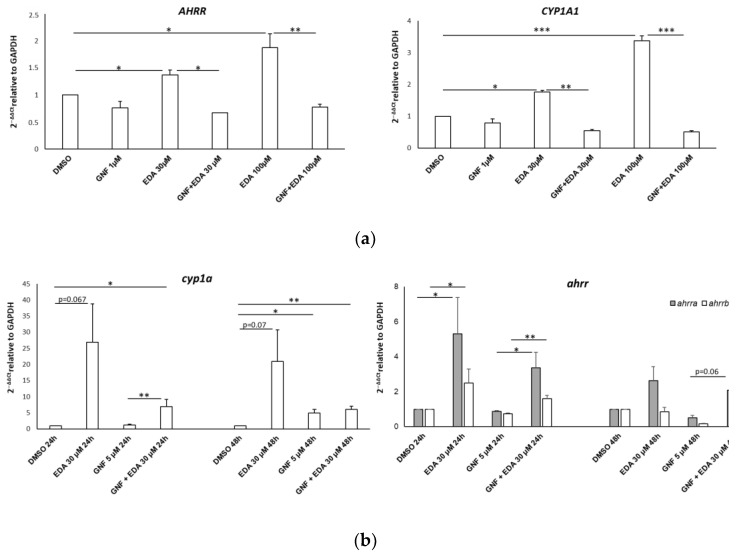
AHR inhibition curtails EDA-mediated AHR target gene upregulation in vitro and in vivo. Bar graphs show the gene expression levels detected by qPCR on RNA obtained from SH-SY5Y cells (**a**) and zebrafish larvae (**b**). Cells were treated with DMSO, 30 µM or 100 µM EDA and/or 1 µM GNF-351 for 24 h. (**b**) Zebrafish larvae at 8 hpf were treated with DMSO or 30 µM EDA in the presence or absence of 1 µM GNF-351 for 24 h and 48 h. The mean ± SEM of 3 experiments is shown. * *p* < 0.05, ** *p* < 0.01 and *** *p* < 0.001 with unpaired Student’s *t*-test. The image was edited using BioRender.com (https://www.biorender.com/ accessed on 3 April 2024).

**Table 1 biomolecules-14-00443-t001:** Biological pathways most significantly modulated by Edaravone treatment in OPCs.

	Category	Term	Count	%	*p*-Value
UP-REGULATED	REACTOME_PATHWAY	Cytochrome P450—arranged by substrate type	3	5.5	1.1 × 10^−2^
	REACTOME_PATHWAY	Synthesis of epoxy (EET) and dihydroxyeicosatrienoic acids (DHET)	2	3.6	1.5 × 10^−2^
	REACTOME_PATHWAY	Synthesis of (16-20)-hydroxyeicosatetraenoic acids (HETE)	2	3.6	1.8 × 10^−2^
	REACTOME_PATHWAY	Phase I—Functionalization of compounds	3	5.5	2.3 × 10^−2^
DOWN-REGULATED	KEGG_PATHWAY	Phagosome	5	2.9	4.7 × 10^−2^
	KEGG_PATHWAY	PI3K-Akt signaling pathway	7	4.1	5 × 10^−2^
	REACTOME_PATHWAY	Mitotic prometaphase	8	4.7	1.3 × 10^−3^
	REACTOME_PATHWAY	Metabolism of water-soluble vitamins and cofactors	5	2.9	1.4 × 10^−2^
	REACTOME_PATHWAY	Metabolism of vitamins and cofactors	6	3.5	1.5 × 10^−2^
	REACTOME_PATHWAY	Nucleotide catabolism	3	1.7	3.8 × 10^−2^
	REACTOME_PATHWAY	Organelle biogenesis and maintenance	6	3.5	4.5 × 10^−2^
	REACTOME_PATHWAY	M Phase	8	4.7	4.9 × 10^−2^
	WIKIPATHWAYS	Translation factors	4	2.3	6.4 × 10^−3^
	WIKIPATHWAYS	Focal adhesion: PI3K-Akt-mTOR signaling pathway	7	4.1	3.7 × 10^−2^

**Table 2 biomolecules-14-00443-t002:** Scores obtained from the different docking tools.

Compound	XP-GScore Glide XP kcal/mol	MMGBSA_dGbind Prime kcal/mol	Binding Energy (BE) Autodock kcal/mol
Indirubin	−11.33	−67.57	−9.08
Leflunomide	−9.074	−51.34	−7.17
Edaravone	−7.55	−45.03	−5.97

## Data Availability

All the data and simulations supporting the findings of this study are available from the corresponding author upon reasonable request.

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
