# Peer review of "The Antioxidant Drug Edaravone Binds to the Aryl Hydrocarbon Receptor (AHR) and Promotes the Downstream Signaling Pathway Activation"

_biomolecules, 2024, doi:10.3390/biom14040443_

Round 1

Reviewer 1 Report

Comments and Suggestions for Authors

The authors performed a methodologically and conceptually sound study. The results are presented and discussed adequately. I have no comments that would improve the manuscript. Thus, I recommend acceptance of the manuscript in its present form.

Reviewer 2 Report

Comments and Suggestions for Authors

In the present manuscript entitled “The antioxidant drug edaravone binds to the aryl hydrocarbon receptor (AHR) and promotes the downstream signaling pathway activation”, Veroni et al. revealed that the aryl hydrocarbon receptor (AHR) as a target of the small molecule Edaravone (EDA) using a transcriptomic strategy. They found that EDA increases the expression of Ahrr, Cyp1a1, and Cyp1b1, the key targets of the AHR pathway in OPCs. Then authors showed that, using SH-SY5Y neuroblastoma cells as a convenient cellular model, EDA induces the translocation of AHR from cytosol to nuclei, upregulating the downstream effectors. In addition, they investigated the EDA activity on the AHR pathway using in vivo zebrafish model. The authors provide interesting mechanism of EDA. Also, the data supporting these conclusions are generally convincing. Following are few concerns for this study.

1)    In Figure 4, there are two SHR bands both in the cytoplasm and in the nucleus. Are either of them the real AHR band? Is one a modified AHR? Also, the band especially in the cytoplasmic 6 hours after EDA treatment are shifted upward. Why is this?

2)    In transcriptome analysis, the authors used oligodendrocytes to identify three genes that act downstream of AHR and have since used SH-SY5Y cells to analyze nuclear translocation of AHR. I know that SH-SY5Y cells are a very useful tool for investigating AHR signals, but why not use oligodendrocyte cell lines? It is necessary to confirm that EDA binds to AHR and activates its downstream signals in oligodendrocytes as well.

3)    In Figure 5, the authors performed an immunoblotting analysis of CYP1A1. It would be more reliable if you include the blots of AHRR. Additionally, since SHSY cells are treated with EDA for 24 hours, are there any cells that are undergoing cell death?

4)    In Figure 6, there was no significant difference in GFP activation between DMSO and EDR treated cells. The authors’ conclusion appears to be correct, increasing the number of n or something would be helpful.

Reviewer 3 Report

Comments and Suggestions for Authors

Veroni et al., demonstrated an important study about the antioxidant drug edaravone. Importantly, they confirmed the results not only in a second cell model but also used zebrafish as in vivo model. The  manuscript is well written. Sometimes the presentation of results can be improved.

(I) Major comment

(A) The western blots demonstrated have mostly problems with the loading control. Often, the samples which is the most important one in the whole blot has a lower or higher loading control compared to the other samples. Normalization through the lower/ higher loading control leads to an overinterpretation of the results. Therefore, better western blots with better loading controls need to be presented for all blots in figure 6, and figure 7.

(B) In each western blot of figure 4 detected with AHR two protein bands of different size are presented. Can the author please comment, which size AHR have, are these different isoforms and which band was quantified in the quantification?

(C) In the paragraph 3.5 the author wrote that they evaluated the target genes cyp1a and ahrr in zebrafish. In the respective figure 8b only cyp1a is shown. Why did the author not present ahrr as written in the main text. This result needs to be integrated in figure 8b.

(D) The author mentioned in line 416/417 that treatment lead to increased GFP expression (Figure S2). In figure S2 a higher total GFP expression is NOT visible. Some larvae represent green puncta which are not visible in untreated larvae. Therefore, can the author rewrite this paragraph and explain what these puncta can be. Additionally, from the three shown treated larvae only two of them show these puncta. Therefore, a quantification of larvae with and without puncta (with and without the drug) need to be integrated to understand how often this phenomenon occur after treatment.

Minor comment

(a) In the material/method section (2.3) the author wrote that the OPCs were treated for 16 hours. In all results, the author commented on 14 h treatment. Here is unclear how long the treatment was performed and therefore, which specification is correct

(b)  Some figure and figure legend are very small and blurred. Therefore, it was not possible to zoom in the pdf document to read the respective figure legend. The is true for e.g. figure 1, figure 2, figure 6 quantification in c and d

Comments on the Quality of English Language

The writing style is fine. But there are several space errors in combination with units (h, µM).

Reviewer 4 Report

Comments and Suggestions for Authors

The manuscript titled “The antioxidant drug edaravone binds to the aryl hydrocarbon receptor (AHR) and promotes the downstream signaling pathway activation” combined in vitro OPC culture and RNA-seq to investigate the potential molecular mechanism of EDA treatment. The results suggested that AHR pathway is altered in the SH-SY5Y cell line and zebrafish embryos. Furthermore, the GFP which linked with Olig2 expression is increased. Although the manuscript is very interesting and with the potential translational novelty, further experiments is necessary for the publication. Here are the comments.

1. The major concerns are coming from the RNAseq of OPC. The early publication in 2018 suggested that Edaravone protected OPCs against oxidative stress in vitro while didn’t cause the OPC cell death. The question is about whether in the normal condition, the EDA work on the same mechanism as under oxidative stress? Design a condition of oxidative stress may solid the ARH as the downstream target of EDA.  

2. While the authors focused on the AHR pathway, the RNA-seq also suggested that mTOR pathway was significantly change. As the major pathway drive the protein synthesis and cell growth, what’s the level of mTOR and its related protein change like AKT, pS6 and their phosphorylation? If it may apply, whether AHR and mTOR pathway are correlated in the authors’ model since there were some hints from published reference.

3. It’s unclear whether the change of NRF2 and olig2 alteration were also observed in the OPC cell culture. This is quite important since later experiments of pathway studies were out of the oligodendrocyte scope.

4. Although the GFP expression is link to the Olig2, it better to measure directly through Olig2 antibody which is more convincible.

5. The results indicated nuclear translocation of AHR and subsequent expression of endogenous AHR target genes. However, if the total AHR expression changed is remaining unclear.

6. in the INDI positive control, the mRNA is dramatically increased but not the protein level. Can authors provide a potential explanation?

Round 2

Reviewer 3 Report

Comments and Suggestions for Authors

All raised concerns were answered and corrected accordingly.

Author Response

We thank the reviewer for her/his valuable comments and observations.

Reviewer 4 Report

Comments and Suggestions for Authors

I have one more concern about the result. The statistic method used in the paper is paired student t-test. However, paired test is used for the data collected from same sample, for example, comparison before and after treatment. The western blot is coming from different independent samples which can't use paired test. This point may change the conclusion of result. Authors need to carefully check it. 

Author Response

Manuscript ID: Biomolecules-2853571

Q1: I have one more concern about the result. The statistic method used in the paper is paired student t-test. However, paired test is used for the data collected from same sample, for example, comparison before and after treatment. The western blot is coming from different independent samples which can't use paired test. This point may change the conclusion of result. Authors need to carefully check it. 

RE: We agree with the reviewer that the unpaired t-test is more appropriate for our analysis. Data relative to Figures 2 and 5 were reanalysed using the unpaired t-test and the figure legends were modified accordingly. The statistical analysis of the zebrafish data had initially been performed using the unpaired t-test, but erroneously stated as paired in the manuscript. The legends of Figures 6, 7, 8 were updated accordingly. All changes are highlighted in green within the resubmitted manuscript (Biomolecules-2853571_revised_2nd round).

Round 3

Reviewer 4 Report

Comments and Suggestions for Authors

Thank you for the quick reply. Good luck for your future studies.